# The Golden Fig: A Plasmonic Effect Study of Organic-Based Solar Cells

**DOI:** 10.3390/nano12020267

**Published:** 2022-01-14

**Authors:** Jessica Barichello, Paolo Mariani, Fabio Matteocci, Luigi Vesce, Andrea Reale, Aldo Di Carlo, Maurizio Lanza, Gaetano Di Marco, Stefano Polizzi, Giuseppe Calogero

**Affiliations:** 1CHOSE—Centre for Hybrid and Organic Solar Energy, Department of Electronic Engineering, University of Rome ‘‘Tor Vergata’’, 00133 Roma, Italy; paolo.mariani@uniroma2.it (P.M.); fabio.matteocci@uniroma2.it (F.M.); vesce@ing.uniroma2.it (L.V.); reale@uniroma2.it (A.R.); aldo.dicarlo@ism.cnr.it (A.D.C.); 2ISM-CNR, Istituto di Struttura della Materia, Consiglio Nazionale delle Ricerche, 00133 Roma, Italy; 3CNR-IPCF, Istituto per i Processi Chimico-Fisici, Consiglio Nazionale delle Ricerche, 98158 Messina, Italy; maurizio.lanza@cnr.it (M.L.); dimarco@ipcf.cnr.it (G.D.M.); 4Dipartimento di Scienze Molecolari e Nanosistemi, Università Ca’ Foscari Venezia, 30172 Venezia-Mestre, Italy; polizzi@unive.it

**Keywords:** DSSC, gold nanoparticles, plasmonic effect, natural dye, DSSM

## Abstract

An optimization work on dye-sensitized solar cells (DSSCs) based on both artificial and natural dyes was carried out by a fine synthesis work embedding gold nanoparticles in a TiO_2_ semiconductor and perfecting the TiO_2_ particle sizes of the scattering layer. Noble metal nanostructures are known for the surface plasmon resonance peculiarity that reveals unique properties and has been implemented in several fields such as sensing, photocatalysis, optical antennas and PV devices. By embedding gold nanoparticles in the mesoporous TiO_2_ layer and adding a scattering layer, we were able to boost the power conversion efficiency (PCE) to 10.8%, using an organic ruthenium complex. The same implementation was carried out using a natural dye, betalains, extracted from Sicilian prickly pear. In this case, the conversion efficiency doubled from 1 to 2% (measured at 1 SUN illumination, 100 mW/cm^2^ under solar simulation irradiation). Moreover, we obtained (measured at 0.1 SUN, 10 mW/cm^2^ under blue light LED irradiation) a record efficiency of 15% with the betalain-based dye, paving the way for indoor applications in organic natural devices. Finally, an attempt to scale up the system is shown, and a betalain-based- dye-sensitized solar module (DSSM), with an active area of 43.2 cm^2^ and a PCE of 1.02%, was fabricated for the first time.

## 1. Introduction

The current energy crisis spurs us to employ pollution-free sustainable energy sources. Since solar radiation is the most abundant energy resource available on Earth [1], photovoltaics (PVs) are being considered one of the most promising technologies based on a renewable source. Recently, numerous research groups have invested much effort in testing new 2D/3D materials, innovative device structures and cheap fabrication procedures to enhance power conversion efficiency (PCE) and lower material costs in order to enable PV devices compete with electricity produced from fossil fuels [2,3]. Dye-sensitized solar cells (DSSCs) belong to the so-called third-generation solar cells. Despite their moderate efficiencies as compared to first-generation silicon cells (a certified PCE of 26% for the best silicon cell and a certified PCE of 13% for DSSCs) [4], DSSC technology continues to attract the attention of the scientific community with 1000 published articles in 2020 [5]. DSSCs have gained widespread interest due to their low production costs, simple fabrication and, above all, their tunable optical properties and the weak dependence of PCE on the incidence radiation angle. Moreover, DSSCs have the feature of increasing PCE under low sun radiation intensity and diffuse light conditions when compared to PV technologies based solely on semiconductors [6]. These last peculiarities identify DSSC devices as some of the most suitable PV technologies for indoor application [7,8,9]. Furthermore, many researchers are proposing DSSCs as power supply cells for Internet of Things (IoT) systems [8,9] underlying connections among wireless sensor nodes, consumer electronic devices, wearable devices and smart meters; since the power needed for IoT devices is typically low, DSSC devices can easily supply their energy consumption under indoor conditions.

The DSSC, or Gräzel’s device [10], has a typical sandwich shape where all active layers are typically sanMdwiched between two transparent conductive glasses. The working electrode (WE) is conductive glass where a mesoporous layer of TiO_2_ (or another semiconductor) lies. The TiO_2_ layer is then sensitized by a dye and impregnated with an electrolyte solution (redox mediator) and the device is finally closed with another conductive glass, the counter electrode (CE), where a catalyst is deposited. The dye absorbs the solar radiation and injects excited electrons in the conduction band of the semiconductor; the redox mediator works to regenerate the oxidized dye. The redox mediator regenerates itself by accepting electrons from the catalyst and, in this way, the electrons’ flow continues again and again.

Since the first presentation of the nanostructured titanium dioxide (TiO_2_) DSSC [10], improvements have been carried out for each constituent of this photo-electrochemical device [11] and, in particular, several functional elements have been added to the initial simple structure, with the aim of improving light harvesting and charge transport, decreasing dark current and reducing charge carriers’ recombination. A state-of-the-art cell has around 11% of PCE, with a ruthenium-based dye and an iodine-based electrolyte [12]. The poor light absorption capability of metal oxides (TiO_2_ absorbs light mainly in the UV region) and of the dye molecules (ruthenium-based dye absorbs light in the visible region) [13] is a hindrance to their PCE growth. Incorporation of plasmonic nanoparticles (NPs), such as Ag and Au NPs, into photovoltaics appears to be a highly promising strategy for the improvement of their PCE [14,15]. In fact, NP scattering induces non-Hermitian coupling between the clockwise and counter-clockwise rotating intracavity modes. Some researchers showed that, by straightforwardly tuning the relative position and the volume of the nanoparticles, achirality and strong chiral behaviors of the reflected fields can be observed due to the asymmetric backscattering. This is a counter-intuitive characteristic of chaotic dynamics. They believe that the chiral light chaos is highly feasible in the laboratory. Beyond their fundamental scientific significance, their results will deepen our understanding of non-linear phenomena and chaos in open systems, and may pave the way toward potential applications [16]. Furthermore, NP-mediated chiral light chaos is based on symmetrical second-harmonic generation [17], which reflects the interaction of optical fields with matter inside the anode. Indeed, it has been found that plasmonic NPs absorb light via the collective oscillations of surface electrons, which can be excited and directly injected into the conduction band of semiconductors [15]. Furthermore, the poor light absorption capability of TiO_2_ can also be extended through the enhancement of the photo-absorption cross-section of dye molecules [18]. In addition, the formation of Schottky barriers at the interface of metal oxide and metal NPs can significantly reduce the recombination rate of photo-excited electron–hole pairs [19]. Therefore, plasmonic NPs of different morphologies showing tuned surface plasmon resonance (SPR) characteristics have been tested for the PCE enhancement of DSSCs [18,19,20,21]. Although the incorporation of bare plasmonic metal NPs has improved the PCE of DSSCs, the use of corrosive iodide-based liquid electrolytes results in reduced long-term stability, unless metal NPs are protected [22,23,24]. Therefore, core–shell nanostructures having a noble metal core and metal oxide shell have gained immense attention in various applications, including DSSCs, due to their unique physical properties with high thermal and chemical stability [23,24,25,26,27,28,29,30]. The aim of this work is to investigate the gold plasmonic effect in a natural dye-based device and to compare it to that of an artificial dye.

In the present work, we developed an original scattering layer that comprises nanoparticles of three different sizes (250, 300, 500 nm), and we compared it with a commercial one (NPs of 250 nm). As a reference dye, we used the ruthenium complex N719 (di-tetrabutylammonium cis-bis (isothiocyanato)bis(2,2′-bipyridyl-4,4′-dicarboxylato) ruthenium(II) [12]. The addition on the semiconductor layer of a thin film (5 µm) of TiO_2_ with large NPs (>20 nm) is well known to boost the optical absorption and ensure adequate light trapping in a device [31,32]. The choice of using three NP sizes, to the best of our knowledge, for the first time in the literature, was made to obtain scattering at different light wavelengths. Device optimization by our homemade scattering layer caused an increment of 7% in the current density (J_sc_) and an 8% increment in PCE. After this, core-shell Au@TiO_2_ NPs were incorporated directly into the TiO_2_ nanoparticles to assemble the photo-active layer (Figure 1). This structure boosts the PCE, mainly influencing the open circuit voltage (V_oc_) (+4%), reaching a PCE of 10.85% (+5%). We tested, for the first time in the literature, the effect of core–shell Au@TiO_2_ NPs in a natural dye-based DSSC with betalains from Sicilian prickly pear. In the struggle to cut costs and to use increasingly pollution-free materials, it is of significant importance to investigate the use of natural dyes that are ideal candidates for environmentally friendly solar cells. Natural DSSCs are a widely explored topic because they are non-toxic, low cost, renewable and abundant, but much effort is needed to increase their efficiency that is currently mainly around 3% [33,34]. Here, with the presence of Au@TiO_2_ NPs, the efficiency doubled from 1 to 2%, mainly influenced by V_oc_ and J_sc_. We performed JV measurement at 0.1 SUN (indoor conditions, blue LED irradiation) with the aim of proving the capability of natural dye to produce high PCE under shadow conditions and considering their application in an indoor context: 15% and 64% of conversion efficiencies were obtained with betalains and N719, respectively. As the last step of this work, we present a first attempt to scale up this device. We realized a natural dye-sensitized solar module (DSSM) with an active area of 43.2 cm^2^ and a PCE of 1.02%. As far as we know, this is the largest module based on a natural dye reported in a published work.

## 2. Materials and Methods

### 2.1. Preparation of TiO_2_ Sub-Micrometric Powder and Paste for Scattering

The sub-micrometric TiO_2_ particles (beads) were prepared via a sol–gel process, followed by a solvothermal treatment in an ethanol/water solution (160 °C, 16 h) and calcination (500 °C, 2 h), as described in [35]. Titanium (IV) isopropoxide (TIP) was used as a titanium precursor and hexadecylammine (HAD) acted as a structure-directing agent in the precursor solution, whereas KCl was used to control the monodispersity of the beads by adjusting the ionic strength of the solution. To obtain the final mesoporous, crystalline anatase TiO_2_ beads with sizes of about 300 nm and 500 nm, in the following solution composition of HDA:H_2_O:KCl:ethanol:TIP (molar ratio), were used: 0.5:5:5.5 × 10^−3^:230.3:1.0 (300 nm) and 0.75:7:5.5 × 10^−3^:230.3:1.0 (500 nm). We used, as a reference, a scattering layer purchased from a company (Dyesol, Queanbeyan, Australia) with nanoparticles of 250 nm. We mixed into the Dyesol paste our homemade TiO_2_ nanoparticles of 300 and 500 nm, in order to have a scattering layer paste with nanoparticles of three sizes: 250, 300 and 500 nm.

### 2.2. Preparation of Au@TiO_2_ Nanoparticles

The synthesis procedure is a combination of (a) a redox reaction carried out by NaBH_4_ assisted by trisodium citrate (TC) as a chelant [36,37], to produce the gold nanoparticles (Au NPs), followed by (b) their functionalization to provide anchoring molecules (modification of ref. [37]), and, finally, (c) the same acid hydrolysis with solvothermal treatment described above to obtain TiO_2_ nanopowders was then used to embed the Au NPs [38,39].

The detailed procedure follows below:(a)The Au NPs are synthesized by adding 1 mL of 11.2 mM NaBH4 dropwise under vigorous stirring to an aqueous solution of 20 mL of 1 mM HAuCl4 and 1.6 mL of 38.8 mM TC, the latter working as a capping agent, which restricts crystal growth to the desired size. By the addition of NaBH4, the colorless solution immediately turns ruby red, indicating the formation of the Au NPs. The sol is stable for about one month, after which some black precipitate starts to form.(b)The TC capping agent is substituted by sodium 3-mercaptopropionate (NaMP), by adding it dropwise, under stirring and at room temperature, to the solution containing the Au NPs. Dropping must be carried out very slowly (1 drop every 20 s), otherwise the solution turns a violet color, indicating that the particles have increased in size.(c)To embed the Au NPs into TiO2 nanoparticles, 80 μL of titanium tetrabutooxide (TTB) is dissolved in 100 mL of ethanol. Then, 50 mL of the solution containing the NaMP-capped Au NPs is added dropwise and, finally, it is left in reflux for 45 min at 80 °C. The dispersion becomes turbid and assumes a violet color. The solution is finally centrifuged 3 times at 9000 rpm for 30 min and the particles are thermally treated in an oven at 500 °C for 30 min.

### 2.3. Preparation of the TiO_2_ Paste

Gold nanoclusters were dissolved in ethanol for a night by stirring; then, they were embedded directly on the 18 NRT TiO_2_ paste (0.7% in weight of Au@TiO_2_) furnished by 18-NRT a company (Dyesol, Queanbeyan, Australia) and mixed for 2 days, alternating stirring and ultrasonic baths. We reproduced the same composition employed by Kamat [30], who found the best efficiency performances upon loading of 0.7% Au@TiO_2_ nanoparticles into the TiO_2_ paste.

### 2.4. Preparation of Electrodes

A 2 cm × 2 cm FTO glass, 2.2 mm thick, (purchased from Pilkington NSG, Chieti, Italy), 7 Ω/sq resistance, was cut and cleaned. Glass substrates were firstly washed with soap and deionized water in order to remove any powder and then dipped in an ultrasonic bath for 10 min, first in acetone and then in an ethanol solution in order to remove any organic residue. After cleaning procedures, the WE was dipped in a chemical bath at 70 °C (TiCl_4_/H_2_O 40 mM) for 30 min. On the WE surfaces, a layer of TiO_2_ was deposited using the screen printing technique, followed by a heating ramp reaching 500 °C for 30 min. Once the WE was ready, while still warm, it was soaked in the dye overnight, at room temperature in the dark. In the CE, a small hole was drilled with the aim to ensure electrolyte insertion. On the conductive side, a H_2_PtCl_6_ solution (5 mM in isopropanol) was dropped and then heated at 480 °C for 30′ after a slow heating ramp. The cell was sealed by assembling, in a sandwich shape, the CE and the WE, the conductive sides together. The sealing was performed by a hot melt ionomer foil of Surlyn (Solaronix, Aubonne, Switzerland), 25 µm thick, placed between the two glasses, shaped in order to leave the TiO_2_ area free, by the use of a thermos-press.

In accordance with the peculiarity of each utilized dye, we used two different homemade electrolytes:J8* (LiI 0.1 M, I_2_ 0.05 M, MPII 0.6 M, TBP 0.5 M in AN:VN 70:30) for N719 [40];AS8* (LiI 0.8 M, I_2_ 0.05 M in AN:VN 85:15) for natural dye, betalains [40,41];where MPII is 1-methyl-3-propyl imidazolium iodide, TBP is 4-tert-butyl-pyridine, AN is acetonitrile and VN is valeronitrile.

As explained in detail in a previous paper [40], the addition of basic compounds such as TBP causes a partial desorption of natural dyes from the TiO_2_ surface and a shift of the absorption spectra towards the red region. This last effect leads to a decrement in the molar extinction coefficient of the natural dye and a general decrease in device performance. On the other hand, the presence of TBP in a ruthenium dye DSSC influences the conductive band of TiO_2_, increasing the V_oc_ and all the photovoltaic parameters [40].

### 2.5. Realization of the DSSC Module

Taking into account the cell fabrication shown above, a similar procedure was applied for the realization of a Z-type DSSC module; two conductive glasses covered with a fluorine-doped tin-oxide (FTO, 7 ohm/square) were cut with the dimensions of 10.5 cm × 9.5 cm. Among them, 8 cells were isolated, etching the FTO with a CO_2_ laser system. The conductive glasses were cleaned with deionized water and soap and then rinsed with acetone and ethanol. Since the screen printing technique is considered one of the most reproducible methods for scaling up to large areas (>10 cm^2^) [42], all active layers were deposited by an automated screen printer. A silver paste (7710 from Chimet, Arezzo, Italy) was screen printed on each electrode (WE and CE) to create vertical connections for each cell in the module. On the WE, a TiO_2_ paste (18 NRT by GreatCell Solar, Rome, Italy) was deposited. We then sintered the titania layer in an oven for 30 min at 500 °C. On the counter electrode, a layer of platinum paste (from 3D Nano, Kraków, Poland) thinner than 1 µm was printed and then fired at 480 °C for 30 min. After dipping the WE in the dye solution overnight (around 16 h), the two electrodes were sealed with a thermoplastic foil (Bynel 60 by DuPont, Milan, Italy) shaped in order to protect the grids and the substrates were sealed by a thermal press. The electrolyte was inserted by the vacuum back filling technique through little channels (around 0.8 mm) left in the Bynel mask. At the end, we sealed the device with a UV-curable commercial resin (ThreeBond^®^, Saint Ouen L’Aumone, France).

### 2.6. Preparation of Dye Sensitizer

A ruthenium complex, N719 (purchased from Solaronix, Aubonne, Switzerland) was dissolved with a concentration of 0.3 mM in ethanol solvent. One hundred grams of Sicilian prickly pear was collected from the plant when fruits reached the typical red color. Later, fruits were washed with distilled water and left to dry. Using a mortar and a pestle, the fruits were mashed to facilitate the dye extraction process. A doubled amount of acid solution (water and HCl 2M) at pH 2, compared to the obtained liquid, was added and the extract was left in an ultrasonic bath for 20 min. At the end, the extract was filtrated.

### 2.7. Measurement

Anode thickness was measured with a profilometer DektakXT Veeco150 (Bruker, Arcore, Italy). X-ray diffraction (XRD) was carried out using a X’Pert system (Bragg–Brentano parafocusing geometry) (Philips, Malvern Panalytical Ltd., Malvern, UK) with nickel-filtered Cu Kα1 radiation (λ = 0.154184 nm). Transmission electron microscope (TEM) images were taken using a JEOL JEM 3010 (Jeol, Tokyo, Japan) operating at 300 kV. For scanning electron microscopy (SEM) images, a SIGMA VP (Zeiss, Oberkochen, Germany) instrument with a field emission gun (FEG) source, using the in-lens secondary electron detector (Everhart-Thornley), and an FEI model QUANTA FEG 450 (FELMI ZFE, Graz, Austria), were used. We obtain the current–voltage (I-V) curves with a digital Keithley 236 multimeter (Farnell^®^, Milan, Italy) connected to a PC. We utilized a solar simulator (Model LS0100-1000, 300 W Xe-Arc lamp, powered by LSN251 power supply equipped with AM 1.5 filter, 100 mWcm^−2^) (LOT-Oriel, Rome, Italy) to simulate sunlight irradiation and the incident irradiance was measured with a Si-based pyranometer. Absorption spectra were detected by a L20 UV–Vis spectrophotometer (range 180–1100 nm) (Perkin Elmer, Milan, Italy). The electrical performance of the DSCMs was measured under a Class B Sun Simulator Solar Constant 1200 KHS (G2V, Edmonton, AB, Canada) at AM 1.5 G, 100 mWcm^−2^, using a Keithley 2420 (Farnell^®^, Milan, Italy) as a source-meter in ambient conditions, calibrated with a SKS 1110 sensor (Skye Instruments Ltd., Llandrindod Wells, Powys, UK). For indoor light irradiation, we chose a typical light tube, a LIFE LED 9W lamp, to test the DSCs under realistic indoor conditions. We measured the light intensity that strikes the DSC by using a modified piranometer PYRIS (Perkin Elmer, Milan, Italy). As relatively few reports exist that focus on the use of DSCs for indoor light harvesting, no standard indoor light source has been established to date.

## 3. Results

### 3.1. The Scattering Layer

With the aim of studying the scattering light effect in DSSCs and to improve the performance of the device, we wanted to fabricate an original scattering layer that was different from the commercial one (Wer 2.0 from Dyesol, Queanbeyan, Australia). We propose a scattering layer with nanoparticles of different sizes, between 300 and 500 nm, in order to improve the light backscattering at different wavelengths. A SEM image (Figure 2) shows the obtained nanoparticles of different sizes. We tested the homemade scattering layer paste (named “scattering J”) with NPs of three sizes (250, 300 and 500 nm) by comparing it with the commercial one (only 250 nm) called “scattering D”. For each batch, composed of five cells, statistical results are shown in Table 1. The most influenced electrical parameter is the current density (J_sc_) that increases by 7% using the three NP sizes in the scattering layer, confirming their ability to scatter light of different wavelengths. After optimizing the device’s scattering layer, we moved on to the gold nanoparticle fabrication.

### 3.2. Au@TiO_2_

With the aim of further increasing electrical performances (such as V_OC_, J_SC_ and PCE), we synthesized TiO_2_ nanoparticles with embedded gold clusters (Au@TiO_2_). As already mentioned, metal clusters should act as a co-sensitizer of the TiO_2_ mesoscopic layer, through the effect of their surface plasmon resonance (SPR) and its influence on the TiO_2_ electronic structure. Furthermore, the increased SPR-induced light scattering in the visible range should both increase the optical path and reduce the transmission of the incident light. Moreover, this increased SPR-induced light scattering allows an enhancement of the rate of photons captured by the dye molecules and their conversion into charge carriers. Regarding the embedding procedure, the gold surface needs to be supplied with a moiety which anchors on TiO_2_. In order to induce electrostatic repulsion between Au NPs and avoid their agglomeration during cappant substitution, a conjugate salt of MPA, namely sodium 3-mercaptopropionate (NaMP), was used. This method successfully produces a stable dispersion of functionalized Au NPs with the original size. In absorbance spectra of Figure 3A, we compare bare and capped gold nanoparticles to embedded Au@TiO_2_; it emerges that no agglomeration occurs. The SPR width suggests that gold nanoparticles with a narrow distribution of sizes are obtained, however, the average size calculated by using Mie theory (11 nm) is clearly overestimated, which is intrinsic to this methodology for very small particles, since surface effects become increasingly important below 10 nm [43]. Au NPs remained stable in size, without agglomerating, during the embedding procedure as well as after calcination. XRD showed that the embedding procedure led to the formation of amorphous TiO_2_ nanoparticles, which transformed into crystalline anatase TiO_2_ nanoparticles after calcination (Figure 3B). The final Au@TiO_2_ material is shown in the TEM images in Figure 4. Most of the Au nanoparticles are below 5 nm in size and are protected inside the TiO_2_ larger nanoparticles (Figure 4c,d). Furthermore, it is possible to notice that Au NPs are not seen to grow in size or agglomerate during the embedding process (Figure 4b); indeed, the observed red-shift in absorbance spectra between Au NPs and Au@TiO_2_ must be related to readjustment of the electronic structure of the material due to the interaction between gold and TiO_2_.

### 3.3. The Working Electrode

In order to investigate the plasmonic effect of gold NPs when mixed in the TiO_2_ paste, UV absorption spectra of solid bulk WEs (FTO/TiO_2_) were made. In Figure 5, we compare the reference TiO_2_ layer with the one mixed with Au@TiO_2_ nanoparticles. The presence of gold can be clearly recognized thanks to the absorbance increment in the spectral region 450–600 nm. This boosting effect of the Au NPs’ SPR has been confirmed on samples impregnated with the N719 dye (notice the large increase in the absorbance peak in Figure 6A), similar to previous work [43,44,45]. However, a different behavior was observed in samples impregnated with the betalain-based dye. Surprisingly, gold nanoparticles do not show the plasmonic effect and do not influence the absorbance intensity, resulting in no differences at all between the reference TiO_2_ and the doped TiO_2_ (Figure 6B). The different molar extinction coefficients (ε) are likely to be the cause of the dissimilar behavior. Indeed, N719 has a lower ε than betalains (ε N719: 1.4 × 10^−4^, ε betalains: 6.5 × 10^−4^ M^−1^cm^−1^) [40,46] and the higher ability of the betalains to capture photons could saturate the absorption. Another reason for the lack of the gold plasmon effect could lie in the different absorptions of betalains in a blue region [39,44] compared to N719; this absorption intercepts the light that should be used by the gold NPs for the plasmon effect.

### 3.4. Photovoltaic Performances

In order to identify the effect of gold nanoparticles on the complete device, we prepared 10 samples of N719 devices (five with Au@TiO_2_ and five without). In accordance with the previous spectrophotometric analysis, the best Au@TiO_2_-containing cell shows a PCE increase from 9.9% to 10.8% (Figure 7A). Statistical analysis is shown in Table 2 and it demonstrates that the most affected parameters are the open circuit voltage (+4%) and the current density (+3%). Here, again, the boost of the device performances by adding Au NPs in the TiO_2_ layer is confirmed. The current density improvement of 3% in this work (Table 2) is due to the charge transfer from gold plasmon to TiO_2_ [14,47]. The TiO_2_ absorption increment, as previously shown in the absorption spectra of Figure 5, derives from the plasmonic resonance effect that excites Au NPs, increasing the energetic field on the TiO_2_ surface. This leads to a strong separation between holes and electrons that reduces charge recombination [14]. Since Au NPs on the TiO_2_ surface act as Schottky barriers, reducing charge recombination, Au NPs not only affect the J_sc_, but also positively influence V_oc_ (+4%, Table 2). This time, the same trend in terms of PCE has been revealed for the gold-containing device based on betalains that doubled the efficiency, from 1.1% to 2% (Figure 7B). Although the absorption spectrum in Figure 6B did not show the resonance effect of the Au NPs in the betalain–gold-based device, the photovoltaic performance shows that gold nanoparticles indeed affect its electrical parameters; both V_oc_ and J_sc_ increase, while a slight decrease in FF is observed. This means that with betalain-based dye, an actual optical effect is not visible for the reasons already explained in Section 3.3, but the electric effect is present. Au NPs reduce charge recombination and, for this reason, the J_sc_ is boosted by 74%, from 6.52 mA/cm^2^ to 11.35 mA/cm^2^ (Table 3). The reason for the 5% enhancement of J_sc_ performances in N719 devices and 74% in betalain devices is explained in reference [48]. Indeed, artificial dyes regenerate in the iodide electrolyte almost four times faster than natural dyes. For this reason, a natural dye benefits, in a wider way, the electronic effect of Au NPs, exponentially improving the J_sc_ performances.

Since DSSC technology provides better performances in terms of PCE [8,9], with a LED lamp set at 0.1 SUN, we performed a JV measurement in order to evaluate the behavior of our devices in indoor environments. The device takes a longer time to recover due to the low photon flux that it receives and the fill factor consequently increases its value. Measurements with an LED lamp also confirmed the precedent behavior; gold nanoparticles boosted the efficiency from 56 to 64% for the N719 dye and from 7.9 to 15% for betalain dye (Table 3).

### 3.5. Natural-Based Dye-Sensitized Solar Module

In order to investigate a real application in the PV market, there is the urgent need for scaling up the DSSC active area (usually of the order of 1 cm^2^ at the lab scale). The dimensions of a DSSC cannot be scaled indefinitely, as there will be a sudden loss of efficiency due to the contact resistance of the TCO and the increasing recombination effects. In general, to scale DSSC technology over a large area (>10 cm^2^), it is necessary to create a device consisting of several cells electrically connected to each other: the DSSC module. In a DSSM, cells connect to each other on the same substrate in parallel (sum of the currents) or in series (sum of the voltages) [49]. The Z-type series, whose interconnections between two adjacent cells are vertical, shows the best DSSM PCE in the literature [50]. In this configuration, the vertical interconnections must be protected by a rugged encapsulation to prevent both the delamination of the device and electrolyte leakage [49]. A demanding engineering procedure is mandatory for module realization [51]. The first attempt to build a natural-based module was carried out as described in Section 2.5 and a PCE of 1.02% was obtained (Figure 8). The reported V_oc_ of 2.79 V means that each cell module has 0.348 V, a higher result in comparison with the single cell (Table 3, Figure 8). The current density of 7.26 mA/cm^2^ is perfectly in accordance with the small cell (Table 3).

## 4. Conclusions

In this work, we compared the plasmonic effect of gold in a DSSC device containing an artificial and a natural dye, respectively. We presented the contribution of the plasmonic effect of gold in a natural-based DSSC. We started with an improvement of the scattering layer effect: adding for the first time in the literature, as far as we know, three different sizes of nanoparticles (250, 300, 500 nm) in the scattering paste that amplifies the optical effect of the scattering layer. In this way, we have demonstrated a J_sc_ and PCE improvement of 7% and 8%, respectively, due to this optical effect. Furthermore, through fine synthesis work, we incorporated gold nanoparticles in a TiO_2_ core–shell to benefit from the plasmonic resonance effect, increasing the PCE of an N719-based DSSC from 9.8 to 10.3% (+5% on average), obtaining a maximum PCE of 10.8%. Once the optimization of the device was completed, we finally tested this structure and the plasmonic effect with a natural dye, betalains extracted from Sicilian prickly pear. We have discovered that the plasmonic gold resonance does not affect the optical properties of a natural-based device, contrary to the artificial one, but the plasmonic contribution is effective from an electrical point of view. In the natural-based device, the optical effect of gold is not visible as the molar extinction coefficient of betalains is much higher than that of N719, thus saturating the absorption effect. On the other hand, in previous work, it was found that an artificial dye regenerates four times faster than a natural one; therefore, the electronic enhancement effect in reducing charge recombination exponentially benefits a natural dye rather than an artificial one. The betalain-based device increased the PCE from 1 to 2% in the presence of gold (1 SUN, solar simulator irradiation). With the aim of indoor applications, we measured both devices under 0.1 SUN (blue LED irradiation) and obtained PCEs of 64% and 15% for the N719 and for the betalains, respectively. In addition, a first attempt was made to manufacture modules with a natural dye to scale up the process. A 42.3 cm^2^ active area module, with a natural-based dye, achieved 1.02% PCE; as far as we know, this is the module based on a natural dye with the largest area reported in the literature.

## Figures and Tables

**Figure 1 nanomaterials-12-00267-f001:**
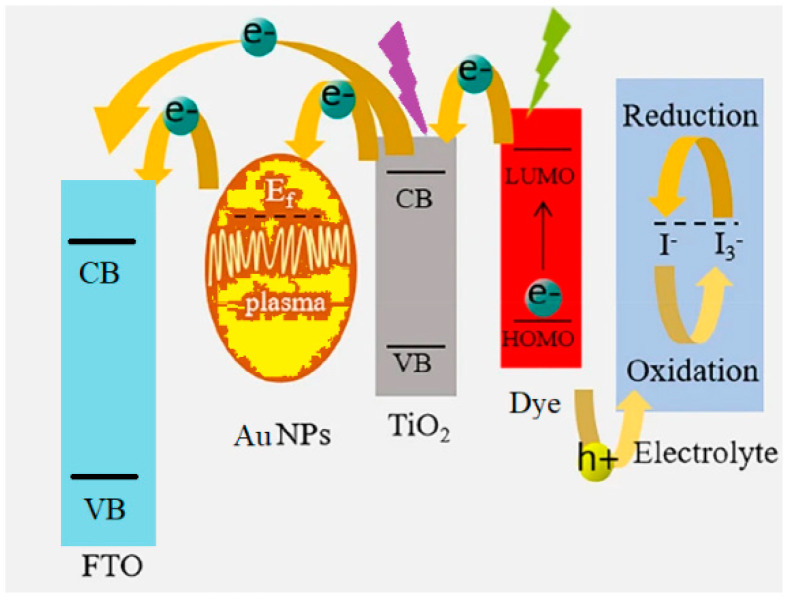
Cross-section view of plasmonic-based dye-sensitized solar cell.

**Figure 2 nanomaterials-12-00267-f002:**
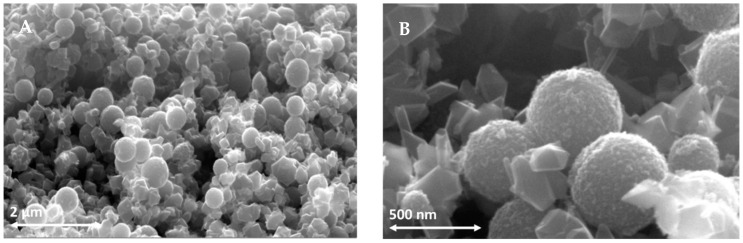
(**A**) SEM image of the sub-micrometric particle sizes (250, 300 and 500 nm) used for scattering layer. (**B**) SEM image of NPs.

**Figure 3 nanomaterials-12-00267-f003:**
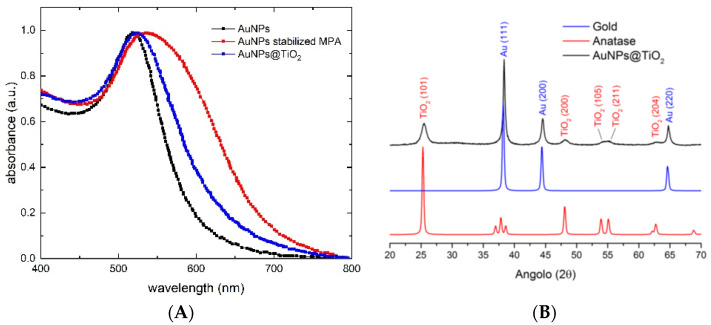
(**A**) Comparison of absorption spectra of bare and capped gold nanoparticles with embedded Au@TiO_2_. (**B**) Comparison of XRD patterns of gold, anatase and Au@TiO_2_.

**Figure 4 nanomaterials-12-00267-f004:**
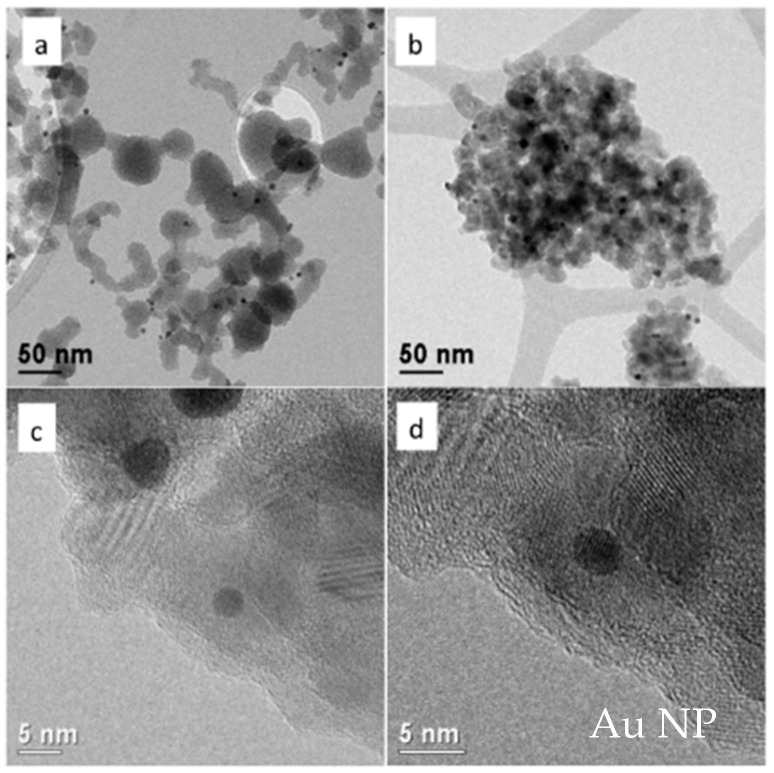
TEM images of the Au@TiO_2_ nanoparticles: (**a**) before calcination and (**b**–**d**) after calcination; (**c**) and (**d**) magnification of a gold nanoparticle (dark spot) surrounded by TiO**_2_** fringes.

**Figure 5 nanomaterials-12-00267-f005:**
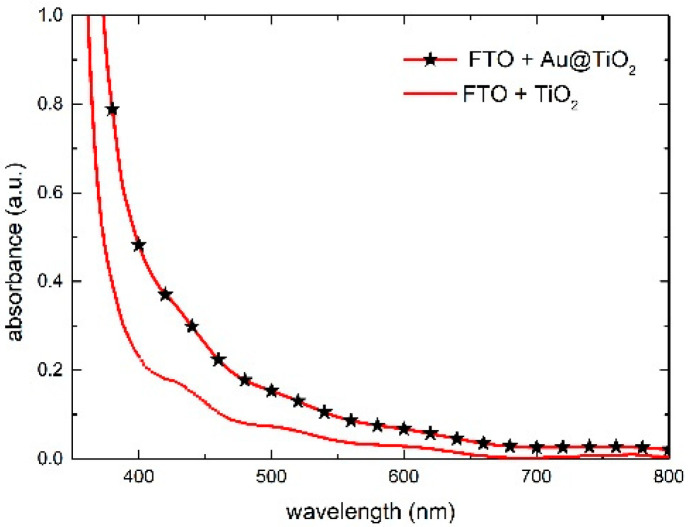
Absorbance spectra of the reference TiO_2_ and the one mixed with Au@TiO_2_.

**Figure 6 nanomaterials-12-00267-f006:**
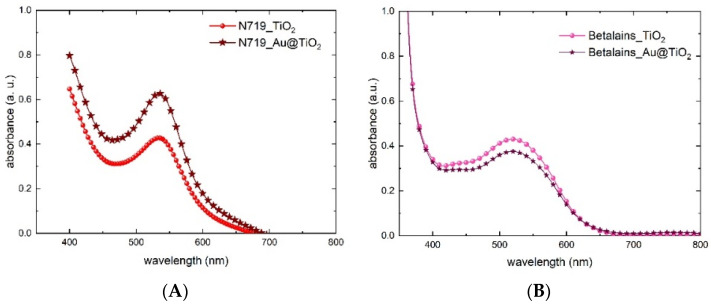
Absorbance spectra of the WE with the reference TiO_2_ and the one mixed with Au@TiO_2_ after dipping in (**A**) N719 and (**B**) betalains.

**Figure 7 nanomaterials-12-00267-f007:**
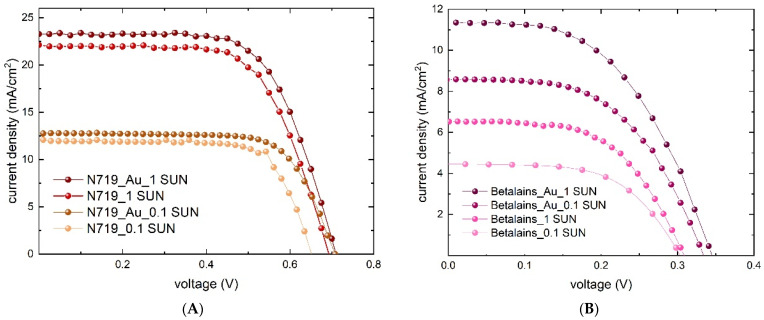
JV measurement of the best DSSC device containing the reference TiO_2_ and the one mixed with Au@TiO_2_ with (**A**) N719 and (**B**) betalains.

**Figure 8 nanomaterials-12-00267-f008:**
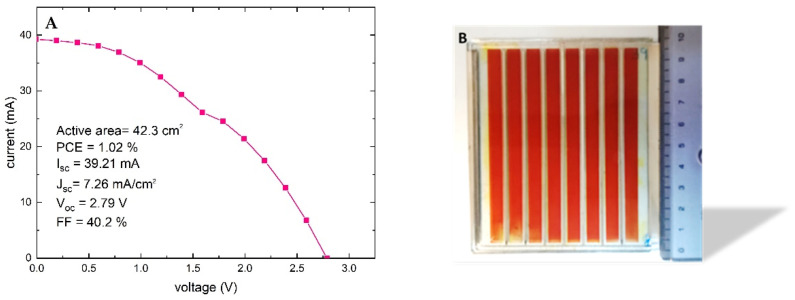
(**A**) I-V curve of the betalain-based DSSM. (**B**) Module sketch.

**Table 1 nanomaterials-12-00267-t001:** Electrical parameter comparison between DSSC devices made with scattering D (Dyesol) and scattering J (homemade)**.** Relative increment is calculated as an average.

	TiO_2_ + scatt. D	TiO_2_ + scatt. J	
J_sc_ (mA/cm²)	20.0 ± 0.7	21.5 ± 0.3	+7%
V_oc_ (V)	0.69 ± 0.01	0.69 ± 0.01	-
FF (%)	64 ± 1	65 ± 1	+1%
η (%)	9.1 ± 0.4	9.8 ± 0.2	+8%

**Table 2 nanomaterials-12-00267-t002:** Photovoltaic parameter comparison between TiO_2_ reference devices and Au@TiO_2_-containing device. Relative increment is calculated as an average.

	TiO_2_@N719	Au@TiO_2_@N719	
J_sc_ (mA/cm²)	21.5 ± 0.3	22.1 ± 0.4	+3%
V_oc_ (V)	0.69 ± 0.01	0.72 ± 0.02	+4%
FF (%)	65 ± 1	65 ± 1	/
η (%)	9.8 ± 0.2	10.3 ± 0.3	+5%

**Table 3 nanomaterials-12-00267-t003:** Electrical parameter at 1 SUN, 100 mW/cm^2^ (solar simulator) and 0.1 SUN, 10 mW/cm^2^ (LED).

Sample	Anode	Irradiation (mW/cm^2^)	J_sc_ (mA/cm^2^)	V_oc_ (V)	FF (%)	PCE (%)
Betalains	TiO_2_@Au	100	11.3	0.34	52	2.0
Betalains	TiO_2_	100	6.5	0.30	57	1.1
Betalains	TiO_2_@Au	10 (LED)	8.6	0.33	54	15.3
Betalains	TiO_2_	10 (LED)	4.4	0.31	57	7.9
N719	TiO_2_@Au	100	23.3	0.71	65	10.8
N719	TiO_2_	100	22.0	0.69	65	9.9
N719	TiO_2_@Au	10 (LED)	12.8	0.71	70	63.6
N719	TiO_2_	10 (LED)	12.0	0.65	72	56.2

## Data Availability

The data presented in this study are available on request from the corresponding authors.

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
