# Peer review of "The Golden Fig: A Plasmonic Effect Study of Organic-Based Solar Cells"

_nanomaterials, 2022, doi:10.3390/nano12020267_

Round 1
Reviewer 1 Report
The papers has serious flaws.
Part 2 and 3.1 ; 3.2 and 3.3 as well as figures must be rewritten and corrected. Some assessments are scientifically untrue (for example fig 5.d : "atomic plane fringes of a gold nanoparticle" : is wrong, fringes observed are from TiO2).
Consider serious scientific evaluation by specialists of the field of gold nanoparticles and nano-materials before resubmitting.
Author Response
We thank reviewer 1 for fruitful suggestions and corrections. Please find revision in the file enclosed, Best regards

Reviewer 2 Report
The manuscript, "The golden fig: a plasmonic effect study of organic-based solar cells” presents a plasmonic study on dye-sensitized solar cells based on both artificial and natural dyes by synthesizing gold nanoparticles, embedding in TiO2 semiconductor (Au@TiO2), and perfecting the TiO2 particle sizes of the scattering layer. The systematic experimental approach, results, and characterization, make this manuscript suitable for publication in Nanomaterials after the following concerns are addressed by the authors prior to its acceptance.
- To be scientifically clear, the manuscript requires certain writing enhancement and grammatical corrections.
- On page 4, line 162: to make Au@TiO2 paste, the authors employed gold nanoclusters that made up 0.7 weight percent of the TiO2. Have the authors tried other weight percents to see how they affect the device performance?
- Several figures can be combined into one as Figure Xa, b, c.
Author Response
We thank Reviewer 2 for fruitful suggestions. Best regards

Reviewer 3 Report
The plasmonic effect of gold is an important frontier in organic-based solar cells. This manuscript compared the plasmonic effect of gold in a DSSC device between an artificial and a natural dye respectively. The authors show that the plasmonic gold effect does not affect the natural-based device by optical effect as in an artificial one, but the plasmonic contribution is from an electrical point of view. The realization of dye sensitized solar cells is important and the idea put forward in this work is new and interesting. The manuscript is well written. The authors may provide some sentences on the nonlinear process, such as Nanoparticle-mediated chiral light chaos based on non-Hermitian mode coupling (Nanoscale, 12(3), 2118-2125) and symmetrical second-harmonic generation (PRA 83, 063845 (2011)), which reflect the interaction of optical fields with matter and may be useful to the readers. The presentation of the results is clear and detailed.
I recommend in favor of this manuscript for publication.
Author Response
We thank Reviewer 3 for the revision. Best regards

Round 2
Reviewer 3 Report
All of my previous questions are very well answered. I recommend publishing this paper at Nanomaterials.